# DEFECT SPECTRUM: A GRANULAR LOOK OF LARGE-SCALE DEFECT DATASETS WITH RICH SEMANTICS

## ABSTRACT

Defect inspection is paramount within the closed-loop manufacturing system. However, existing datasets for defect inspection often lack the precision and semantic granularity required for practical applications. In this paper, we introduce the Defect Spectrum, a comprehensive benchmark that offers precise, semantic-abundant, and large-scale annotations for a wide range of industrial defects. Building on four key industrial benchmarks, our dataset refines existing annotations and introduces rich semantic details, distinguishing multiple defect types within a single image. Furthermore, we introduce Defect-Gen, a two-stage diffusion-based generator designed to create high-quality and diverse defective images, even when working with limited datasets. The synthetic images generated by Defect-Gen significantly enhance the efficacy of defect inspection models. Overall, The Defect Spectrum dataset demonstrates its potential in defect inspection research, offering a solid platform for testing and refining advanced models. Our project page is in https://defect-spectrum-authors.github.io/defect-spectrum/.

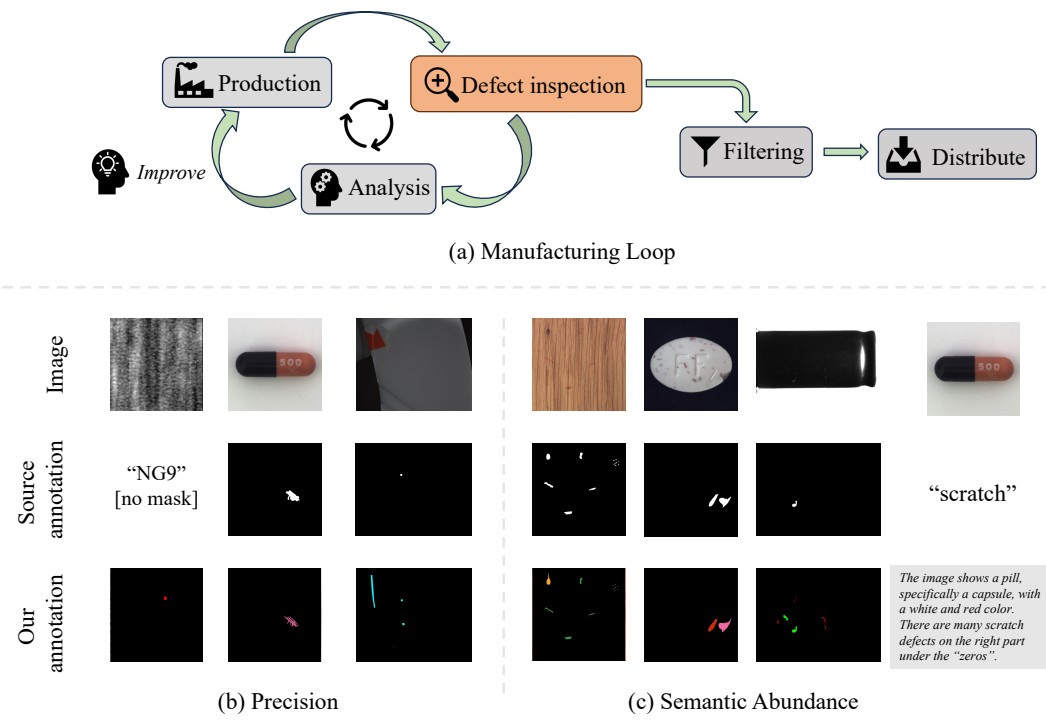

Figure 1: (a) The closed-loop system in industrial manufacturing. Defect inspection plays a pivotal role. (b, c) An overview of our improvements in annotations, in the aspect of precision and semantics abundance. **Best viewed in color.**

# 1 INTRODUCTION

Industrial manufacturing is a cornerstone of modern society. In an environment where minute imperfections can result in significant failures, ensuring top-tier quality is imperative. Manufacturing predominantly relies on a closed-loop system, encompassing production, defect inspection, filtering, and analysis, as illustrated in Figure 1.

Within this system, defect inspection plays a pivotal role, interfacing with most stages and ultimately determining product quality. Striking the right balance between identifying defective items and acknowledging sub-optimal ones, based on defect size, position, and type, becomes critical (Wagner, 2016). For instance, consider a metal plate. If there is a large area on its surface where the paint has peeled off, the functional impact on the metal plate might be minimal despite the significant size of the defect. However, if there s a tiny crack within the metal, even though it is small in size, it might lead to the plate breaking under pressure, significantly affecting its performance. Additionally, documenting the category and location of defects can pave the way for predictive maintenance and provide valuable insights for refining product repair processes (Ni et al., 2022).

However, current datasets struggle to meet the intricate practical needs of industrial defect inspection. One notable limitation is the insufficient granularity concerning defect types and locations. For instance, anomaly detection datasets like MVTEC (Bergmann et al., 2019) and AeBAD (Zhang et al., 2023) give pixel-level annotations but are restricted to binary masks. Meanwhile, datasets like VISION (Bai et al., 2023), though more detailed, occasionally miss or misclassify defect instances.

To address these gaps, we introduce the Defect Spectrum, aiming to offer semantics-abundant, precise, and large-scale annotations for a broad spectrum of industrial defects. This empowers practical defect inspection systems to furnish a more thorough and precise analysis, bolstering automated workflows. Building on four key industrial benchmarks, Defect Spectrum offers enhanced annotations through a rigorous labeling endeavor. We have re-evaluated and refined existing defect annotations to ensure a holistic representation. For example, contours of subtle defects, like scratches and pits, are carefully refined for better precision, and missing defects are carefully filled with the help of specialists. Beyond that, our dataset stands out by providing annotations with rich semantics details, distinguishing multiple defect types even within a single image. Lastly, we have incorporated descriptive captions for each sample, aiming to integrate the use of Vision Language Models (VLMs) in upcoming studies. During this endeavor, we employ our innovative annotation tool, Defect-Click. It has largely accelerated our labeling process, emphasizing its utility and efficiency, ensuring meticulous labeling even with the extensive scope of our dataset.

Another palpable challenge is the limited number of defective samples in datasets. For instance, in DAGM, there are only 900 defective images. In MVTEC, although it has 5354 total images, the defectives among them are merely 1258. And even the extensive VISION dataset falls short in comparison to natural image datasets like ImageNet (Deng et al., 2009) (1 million images) and ADE20k (Zhou et al., 2017; 2019) (20k images). To address this, we harness the power of generative models, proposing the "Defect-Gen", a two-stage diffusion-based generator. Our generator exhibits promising performance in image diversity and quality even with a limited number of training data. We show that these generated data could largely boost the performance of existing models in our Data Spectrum benchmark.

To summarize, our contributions are listed as follows.

- We introduce the Defect Spectrum dataset, designed to enhance defect inspection with its semantics-abundant, precise, and large-scale annotations. Unlike existing datasets, we not only refine existing annotations for a more holistic representation but also introduce rich semantics details. This dataset, building on four key industrial benchmarks, goes beyond binary masks to provide more detailed and precise annotations.

- We propose the Defect-Gen, a two-stage diffusion-based generator, to tackle the challenges associated with the limited availability of defective samples in datasets. This generator is shown to boost the performance of existing models, by enhancing image diversity and quality even with a limited training set.

- We conducted a comprehensive evaluation on our Defect Spectrum dataset, highlighting its versatility and application across various defect inspection challenges. By doing so,

we provide a foundation for researchers to evaluate and develop state-of-the-art models tailored for the intricate needs of industrial defect inspection.

# 2 RELATED WORK

**Industrial Datasets** There are several well-used datasets for Industrial Defect Inspection: DAGM2007(Wieler & Hahn, 2007), AITEX(Silvestre-Blanes et al., 2019), AeBAD(Zhang et al., 2023), BeanTech(Mishra et al., 2021), Cotton-SFDG(Incorporated, 2023) and KoektorSDD (Tabernik et al., 2020) offer commonly seen images that cover a wide array of manufacturing materials; MVTEC (Bergmann et al., 2019; 2021) is a dataset for benchmarking anomaly detection methods with a focus on industrial inspection; VISION V1(Bai et al., 2023) includes a collection of 14 industrial inspection datasets containing multiple objects. A notable shortcoming in the aforementioned industrial datasets is they often lack specificity regarding the defect's type or its precise location. Aiming to refine these issues, we introduce the Defect Spectrum datasets. Further details will be explained in Section 3.

**Defect-mask Generation** Defect inspection plays a vital role in various industries, including manufacturing, healthcare, and transportation. Previous attempts based on the traditional computer vision method (Song et al., 2015) have proven to be robust for detecting small defects, but they all suffer from detecting defects in textures-rich patterns. In recent years, Convolutional Neural Networks(CNNs) (Faghih-Roohi et al., 2016; Mundt et al., 2019; Guo et al., 2021) based models are commonly used for defect inspection, but limited availability of real-world defect samples remains a challenge. To mitigate such data-insufficiency issue, traditional methods for synthesizing defect images manually destroy normal samples (Mery & Filbert, 2002) or adopt Computer-Aided Drawing (CAD) (Mery et al., 2005; Huang et al., 2009). Deep learning-based approaches are generally effective, but they require large amounts of data. GAN-based methods (Niu et al., 2020; Wei et al., 2022; Du et al., 2022; Zhang et al., 2021) are adopted to perform defect sample synthesis for data augmentation. DefectGAN adopts an encoder-decoder structure to synthesize defects by mimicking defacement and restoration processes. However, it is important to note that GAN-based methods typically require a substantial quantity of real defect data in order to achieve effective results. Recent advancements in Diffusion models (Ho et al., 2020; Dhariwal & Nichol, 2021; Nichol & Dhariwal, 2021) demonstrated a superior performance in image generation. However, they tend to reproduce existing samples when trained with scarce data, leading to a lack of diversity. Stable Diffusion (Rombach et al., 2021) is one of the most prevailing methods in this field. Nonetheless, it is not applicable to use a pre-trained stable diffusion model when generating masks. Our proposed approach, on the other hand, is capable of generating defective image-mask pairs with both diversity and high quality, even when trained on limited datasets.

# 3 DATASET

## 3.1 DATASETS ANALYSIS

In Table 1, we present an analysis of the Defect Spectrum datasets in comparison with other prevalent industrial datasets. Notably, the DAGM2007 and Cotton-Fabric datasets originally lacked pixel-wise labels, making them less suitable for detailed defect inspection. While datasets like AITEX, AeBAD, BeanTech, and KoektorSDD offer defect masks, they only focus on a limited range of products, offering a restricted number of annotated images and defect categories.

High-quality datasets such as MVTEC and VISION, despite providing a substantial number of images with pixel-level annotations, still have some limitations. For instance, there are cases where MVTEC and VISION annotations either miss defects or provide imprecise, coarse labels, as illustrated in Figure 1(b). Moreover, these datasets sometimes merge distinct defect classes into a single generic category, evident with the "toothbrush" in MVTEC and the "capacitor" in VISION. In MVTEC, only a binary mask is provided, which assumes each image contains just one defect type. This assumption is contrary to real-world situations where images, including many from MVTEC, often obtain multiple defect types at the same time, as highlighted in Figure 1 (c).

To address these challenges, we introduce the Defect Spectrum datasets, comprising 3518 carefully selected high-quality images sourced from existing industrial datasets, including MVTEC, VISION-V1, DAGM2007, and Cotton-Fabric. Our dataset ensures that every selected image is accompanied by detailed, precise, and diverse category annotations. Furthermore, we provide comprehensive captions to enhance contextual understanding. For each product type in our datasets, we also offer a variety of realistic synthetic data along with their corresponding accurate masks.

Table 1: Comparison with real-world manufacturing datasets. Defect Spectrum datasets are the second largest one even though excluding our synthetic data. Defect Spectrum is also the most diverse, semantics-abundant, and precise manufacturing benchmark datasets to date. We use * to represent the amount of synthetic data.

| | Annotated Defective Images | Defect Type | Pixel-wise Label | Multiple Defective Label | Detailed Caption |
|---|---|---|---|---|---|
| AITEX | 105 | 12 | ✓ | | |
| AeBAD | 346 | 4 | ✓ | | |
| BeanTech | 290 | 3 | ✓ | | |
| Cotton-Fabric | 89 | 1 | | | |
| DAGM2007 | 900 | 6 | | | |
| KolektorSDD2 | 356 | 1 | ✓ | | |
| MVTec | 1258 | 69 | ✓ | | |
| VISION V1 | 4165 | 44 | ✓ | ✓ | |
| Defect Spectrum | 3518+1920* | 125 | ✓ | ✓ | ✓ |

## 3.2 ANNOTATION IMPROVEMENTS

Our improvements in annotations are mainly in three aspects: precision, semantics-abundance, and detailed caption.

**Precision** For datasets that were neither annotated nor had image-wise annotations, we have elevated them to meet our standards. We have enriched these datasets with meticulous pixel-level annotations, delineating defect boundaries and assigning a distinct class label to each type of defect. For those datasets that already possessed pixel-wise masks, we enhanced their precision and rectified any imperfections. We undertook efforts to account for any overlooked defects, ensuring exhaustive coverage. For nuanced defects, such as scratches and pits, we refined the contours to achieve heightened accuracy.

**Semantics Abundance** In contrast to datasets that only offer binary defective masks, Defect Spectrum furnishes annotations with more semantic details, identifying multiple defect types within a single image. We identify that there are 552 multiple defective images and provide their multi-class labels. Moreover, we have re-assessed and fine-tuned the existing defect classes, guaranteeing a more granular and precise categorization. In total, we offer 125 distinct defect classes.

**Detailed Caption** With the evolution of Vision Language Models (VLMs), we have equipped our datasets by integrating exhaustive captions. It's worth noting that current captioning models, such as BLIP2 (Li et al., 2023) and LLaVa (Liu et al., 2023), often overlook defect information. As a remedy, we conducted manual inspections of each defective image and furnished detailed descriptions. These narratives not only identify the objects but also elucidate their specific defects. We anticipate that this enhancement will inspire researchers to increasingly leverage VLMs for defect inspection in forthcoming studies.

## 3.3 AUXILIARY ANNOTATION TOOL

Annotating pixel masks is an exceptionally demanding task in the labeling domain, especially under the stringent standards of Defect Spectrum. It is not feasible to perform such a task from scratch. To alleviate this challenge, we introduce an auxiliary annotation tool, "Defect-Click," designed to conserve the efforts of our specialists.

Source image
Segment Anything

Defect Click

Figure 2: Annotating a scratched capsule: With the "Defect-Click" tool, we can swiftly pinpoint the two scratches. However, when using "Segment Anything", it becomes challenging to accurately identify the defects. **Best viewed in color.**

Defect-Click is an advanced interactive annotation tool designed to automatically segment defect areas based on a user's click point. Distinct from traditional interactive segmentation methods, Defect-Click utilizes its pretrained knowledge of industrial defects to adeptly pinpoint irregular defective regions. Built upon the Focal-Click framework (Chen et al., 2021; 2022), we tailored Defect-Click for the industrial defect domain by integrating 21 proprietary labeled datasets, introducing multi-level crop training for small defects, and incorporating edge-sensitive losses during training. These specialized approaches ensure that Defect-Click significantly outperforms other annotation tools in the industrial dataset domain, as showcased in Figure 2. Segment Anything (Kirillov et al., 2023) struggles to identify the scratch defect, while Defect-Click clearly delineates the defect's contour.

With the assistance of Defect-Click, we can initially obtain a rough defect mask with merely several clicks and subsequently refine its imperfections. On average, this approach has resulted in a time-saving of about 60%. Even though, this comprehensive annotation project still spans a total of 580 working hours.

## 3.4 DEFECT GENERATION

To tackle the issue of defects scarcity, we turn to the burgeoning field of generative models. By using the limited available data, we propose a two-staged diffusion-based generator, called the "Defect-Gen".

### 3.4.1 BACKGROUND

Given a set of defective image-mask pairs, we aim to learn a generative model that captures the true data distribution, so that it can generate more pairs to augment the training set. We denote the dataset as $\mathcal{D} = \{(I_1, M_1), (I_2, M_2), \ldots, (I_N, M_N)\}$, where $I_i \in \mathbb{R}^{h \times w \times 3}$ and $M_i \in \{0, n_{defect}\}^{h \times w \times n_{defect}}$ refer to the defect image and its corresponding defect mask respectively. $N$ is the number of samples in the training set, which is small in practice. $n_{defect}$ denotes the number of defective types in the mask images. Specifically, we convert the mask into a one-hot encoding scheme for each channel separately. We show that with a very small modification, it can generate images with corresponding labels. We perform a channel-wise concatenation between $I$ with the $M$, i.e., $x = I \oplus M$, where $\oplus$ means concatenation and $x \in \mathbb{R}^{h \times w \times n_{total}}$, and $n_{total} = n_{defect} + 3$. We then treat the $x$ as the *input* to train the generator. This improves the usability of the generative model with negligible computational overhead. In the following, we term $x$ as "image" instead of "image with label" for convenience.

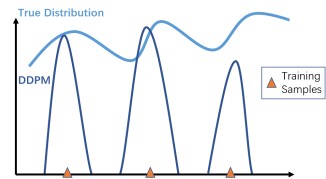

Figure 3: DDPM predicts high density around training samples and fails to capture the true data distribution.

**Few-shot Challenges** Note that defect images are difficult to collect in practice, and thus, models have to be trained with very few samples. Under this situation, we observe that the generated results lack diversity. To be specific, models tend to memorize the training set. The reason could be that the generative models such as Diffusion models tend to predict high density around training samples and fail to capture the true data distribution, as depicted in Figure 3.

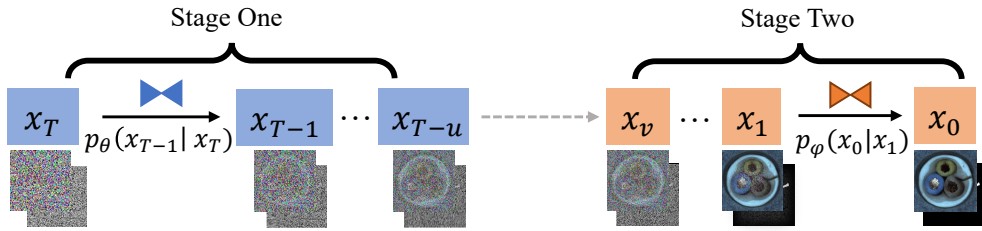

Figure 4: The inference process of the two staged diffusion models. The input to the large model $p_\theta$ is gaussian noise, after the optimal step is reached, the intermediate results containing global information will be used as the input to the small model $p_\phi$.

### 3.4.2 DEFECT-GEN

The limitation discussed above is not surprising. In statistical learning theory, it is well-known that the generalization capacity of a classification model is positively related to the sample size and negatively related to the dimension. We can reasonably hypothesize that a similar trend also holds in the diffusion model according to the Vapnik–Chervonenkis theory (Vapnik & Chervonenkis, 2015). In this sense, as the data dimension ($h \times w \times n_{total}$) is much larger than the sample size ($N = 25$ in our setting), it is not surprising that the vanilla model suffers severe overfitting.

**Modeling the Patch-level Distribution** To alleviate the aforementioned problem, we propose to model the patch-level distribution instead of the image-level distribution. By treating a patch as one sample, the data dimension ($h_{patch} \times w_{patch} \times n_{total}$) is largely reduced, while the sample size ($N_{patch}$) is significantly increased. This reduces the risk of overfitting.

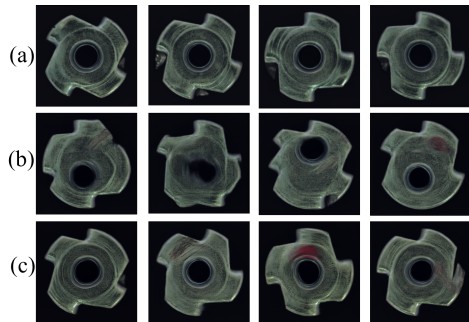

Figure 5: The visual cases in (a) demonstrate a lack of diversity in using DDPMs. Cases in (b) demonstrated excessive diversity. (c) shows the generated samples using our framework. we maintained the global structure while introducing local variance.

**Restraining the Receptive Field** Although we can naively replace $x$ with cropped image patches to achieve patch-level modeling, it is hard to use learned patches to reconstruct into a whole image during inference. In other words, if explicitly train a patched generator, we would have to introduce a reconstruction term to merge these patches. Alternatively, we leverage the network architecture to restrain the size of the receptive field to achieve this. Standard U-Net is used in the vanilla diffusion model (Ho et al., 2020). It is composed of a series of down-sampling layers. With the reduced number of down-sampling layers, the output receptive fields gradually decrease. This allows the model to only be visible to small patches on the original images. This strategy does not change the position of each patch in one image and thus has the potential to maintain the whole image. Thus, by using a smaller receptive field, patch-level modeling is achieved. The detailed size selection of the smaller receptive field will be elaborated in the supplementary section.

**Handling the Global Distortion** While patch-level modeling is effective in overcoming overfitting, it falls short of representing the global structure of the entire image, leading to unrealistic results. This is shown in Figure 5(b). To address this issue, we propose a two-stage diffusion process as depicted in Figure 4. Our approach is inspired by Choi et al. (2022), which reveals that different time steps in the diffusion process correspond to distinct levels of information. In the early stages, coarse geometry information is generated, while in later stages, finer information is produced.

Specifically, we train two models: one with a small receptive field, which we introduced previously, and another with a larger receptive field. During inference, we use the large-receptive-field model to capture the geometry structure in the early steps, and then switch to the small-receptive-field model to generate diverse local patches in the remaining steps. This introduces a hyperparameter, the

switch timestep $u$, and the selection of the timestep will be further discussed in the supplementary section.

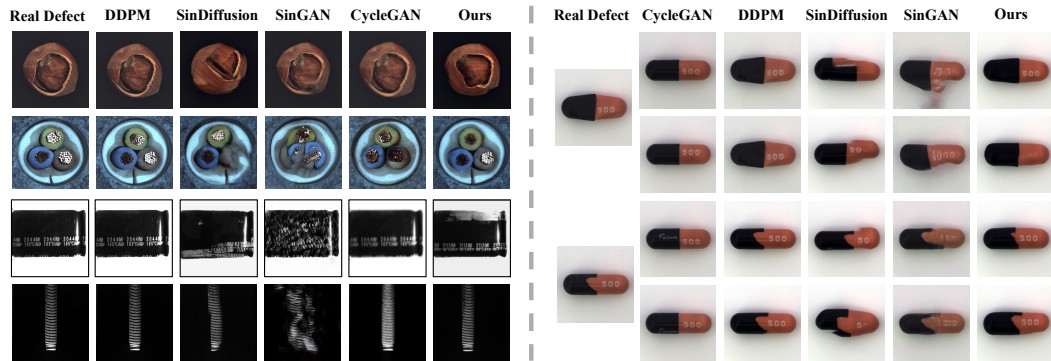

Figure 6: Qualitative comparison of our method with other image synthesis methods. We compared different objects across different datasets to demonstrate the high fidelity of our generation method. On the right-hand side, we show our method can exhibit diversity while maintaining high quality. **Best viewed in color.**

## 4 EXPERIMENTS

### 4.1 BASELINES

In the realm of industrial defect inspection, there are three primary tasks: defect detection (determining if an image contains a defect), defect classification (identifying the type of defect), and defect segmentation (pinpointing both the boundaries and the type of the defect in the image) (Carvalho et al., 2022). Typical defect detection methods such as Patchcore (Roth et al., 2022), PADIM (Defard et al., 2020), and BGAD Yao et al. (2023) emphasize identifying the presence of defects but fall short in discerning defect types. Defect classification methods can determine the type of defect but do not provide information about its location or size. Our Defect Spectrum dataset come with detailed and comprehensive annotations, aiming to solve the most complex task. Consequently, we focus on methods that excel in defect segmentation.

Additionally, due to the confidential nature of many industrial products, transferring data externally is often prohibited. This necessitates models that can operate efficiently on local devices. With this in mind, we have handpicked several SOTA segmentation methods and adapted them to a lightweight version. Our baseline includes UNet-small (Ronneberger et al., 2015), ResNet18 (He et al., 2016)-PSPNet (Zhao et al., 2017), ResNet18-DeepLabV3+ (Chen et al., 2018), HRNetV2W18-small (Wang et al., 2020), BiseNetV2 (Yu et al., 2021), ViT-Tiny (Dosovitskiy et al., 2021)-Segmenter (Strudel et al., 2021), Segformer-MiT-B0 (Xie et al., 2021), and HRNet-Mask2Former (Cheng et al., 2022). The models are abbreviated as follows: UNet (UNet-small), PSP (ResNet18-PSPNet), DL (ResNet18-DeepLabV3+), HR (HRNetw18small), Bise (BiseNetV2), V-T (ViT-Tiny+Segmenter), M-B0 (Segformer-MiT-B0), and M2F (HRNet-Mask2Former). We evaluate the inference time of these baselines on a NVIDIA RTX 3090. Results are presented in Table 2.

Table 2: Speed Evaluation of the baselines. Inf. time denotes the inference time of a single image on NVIDIA RTX 3090.

|                | UNet | PSP  | DL   | HR   | Bise | V-T  | M-B0 | M2F  |
| -------------- | ---- | ---- | ---- | ---- | ---- | ---- | ---- | ---- |
| inf. time (ms) | 33.9 | 26.2 | 33.0 | 15.7 | 23.5 | 38.7 | 17.9 | 68.2 |

### 4.2 EVALUATION OF BASELINES

We present a comprehensive evaluation of the above methods on each sub-set of our Defect Spectrum benchmark. For the performance metric, we choose the mean Intersection over Union (mIoU).

Results are shown in Table 3. The consistent performance of DeepLabV3+ across multiple datasets suggests that it's a robust model for various types of defect segmentation tasks. The Transformer-based models seem to be particularly effective for Cotton-Fabric. This might be due to the inherent advantages of Transformers in capturing long-range semantic information, which could be commonly found in "Cotton-Fabric". Performance varies across models for different categories, suggesting no universal solution. Model selection should consider dataset specifics. Some datasets challenge all models, highlighting a need for more research.

Table 3: Quantitative comparison of various defect segmentation methods across different Defect Spectrum reannotated datasets. Results reflect the mIoU. We highlight the best mIoU of each dataset with red color. "DS" is abbreviated for Defect Spectrum.

| | | CNNs | | | | | Transformers | | |
|---|---|---|---|---|---|---|---|---|---|
| | | UNet | PSP | DL | HR | Bise | V-T | M-B0 | M2F |
| | bottle | 43.44 | 50.20 | 56.53 | 45.02 | 44.92 | 69.71 | 40.88 | 53.20 |
| | cable | 47.95 | 52.50 | 52.59 | 50.39 | 45.24 | 54.51 | 58.31 | 49.72 |
| | capsule | 28.05 | 29.59 | 35.49 | 34.02 | 28.30 | 33.94 | 38.95 | 26.91 |
| | carpet | 50.91 | 53.76 | 53.75 | 47.28 | 44.52 | 43.70 | 38.45 | 47.34 |
| | grid | 37.06 | 42.86 | 41.18 | 30.97 | 33.89 | 40.08 | 18.86 | 24.81 |
| | h_nut | 58.84 | 56.87 | 61.78 | 59.31 | 57.53 | 55.07 | 59.60 | 56.72 |
| Defect | leather | 57.56 | 61.42 | 54.56 | 55.45 | 57.89 | 47.85 | 50.80 | 53.96 |
| Spectrum | m_nut | 49.18 | 46.99 | 51.08 | 48.76 | 55.51 | 54.68 | 48.89 | 39.43 |
| (MVTec) | pill | 35.81 | 36.38 | 33.83 | 29.30 | 27.23 | 42.65 | 46.35 | 27.14 |
| | screw | 31.87 | 38.77 | 33.36 | 29.66 | 19.01 | 22.54 | 19.26 | 21.89 |
| | tile | 85.49 | 82.51 | 83.02 | 85.66 | 84.21 | 78.29 | 79.14 | 83.04 |
| | t_brush | 23.96 | 25.25 | 25.16 | 26.25 | 25.58 | 33.30 | 32.22 | 28.26 |
| | tran. | 40.37 | 44.02 | 58.23 | 44.50 | 45.97 | 53.60 | 41.13 | 50.87 |
| | wood | 72.69 | 67.93 | 68.21 | 69.00 | 67.81 | 62.62 | 73.02 | 59.66 |
| | zipper | 54.83 | 60.95 | 58.03 | 55.87 | 47.15 | 51.69 | 60.12 | 49.47 |
| | **mean** | **49.88** | **51.41** | **51.58** | **47.99** | **45.40** | **49.31** | **46.45** | **45.70** |
| | Capa. | 57.04 | 54.01 | 52.75 | 54.56 | 54.36 | 56.30 | 59.29 | 57.27 |
| Defect | Console | 35.32 | 30.77 | 32.67 | 31.70 | 30.45 | 22.48 | 25.64 | 32.50 |
| Spectrum | Ring | 52.37 | 56.16 | 56.97 | 60.17 | 54.06 | 44.27 | 52.21 | 62.09 |
| (VISION) | Screw | 53.13 | 53.91 | 55.20 | 52.46 | 51.76 | 36.87 | 47.54 | 52.05 |
| | Wood | 64.75 | 66.80 | 66.72 | 66.79 | 67.40 | 53.34 | 63.43 | 66.70 |
| | **mean** | **52.52** | **52.33** | **52.86** | **53.14** | **51.61** | **42.65** | **49.62** | **54.12** |
| DS-DAGM2007 | | 85.89 | 85.14 | 86.82 | 84.02 | 83.14 | 52.42 | 83.06 | 85.56 |
| DS-Cotton-Fabric | | 39.03 | 48.73 | 47.55 | 41.13 | 46.82 | 51.29 | 50.52 | 64.09 |

## 4.3 GENERATION QUALITY

Figure 6 provides a qualitative comparison between our generation results and those from other synthesis methods. On the left-hand side, we used the two images shown in the "Real defect" to generate samples. On the right-hand side, we present differ-

Table 4: Performance (mIoU) comparison between models trained with and without synthetic data.

| | MVTec | VISION | Cotton |
|---|---|---|---|
| w/o synthetic | 51.58 | 54.12 | 64.09 |
| w. synthetic | 55.55 | 55.47 | 65.39 |

ent objects to demonstrate the high fidelity of our method. We observe that the generated models from CycleGAN and DDPM completely failed to learn a diverse defect pattern and thus failed to generate samples with diversity by producing mere duplicates of the training set. On the other hand, sinDiffusion (Wang et al., 2022) and SinGAN (Rott Shaham et al., 2019) can produce diverse samples but are not visually realistic. More visual cases, including other classes, can be found in the supplementary file. Figure 7 displays the image-mask pairs we generated. Our images are of high quality, and the corresponding masks align well with them.

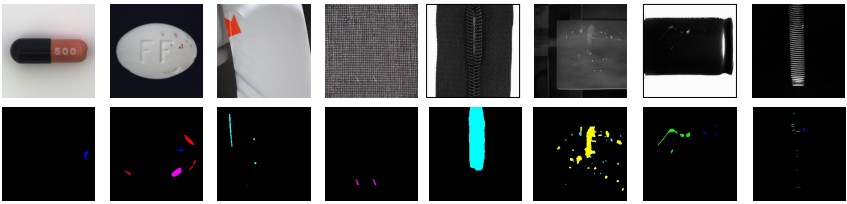

Figure 7: Qualitative comparison of our method with other image synthesis methods. We compared different objects across different datasets to demonstrate the high fidelity of our generation method. On the right-hand side, we show our method can exhibit diversity while maintaining high quality. **Best viewed in color.**

### 4.4 SYNTHETIC DATA FOR PERFORMANCE BOOST

**Boosting SOTA methods with Synthetic Data**   We present the performance improvement when using our synthetic data in Table 4. For each dataset, we report the performance of the best baseline. We do not generate extra data for DAGM2007, since it is already a synthetic dataset. The result demonstrates the effectiveness of our synthetic data. We also compared with other generation methods in the ability to boost performance. Detailed comparisons can be found in the supplementary file.

**Impact of Synthetic Data**   In Figure 8, we delve deeper into the impact of varying the quantity of our synthetic data on model performance. Figure 8 (a) shows the performance improvement over different quantities of synthetic data using DeepLabV3+. Interestingly, we found that the transformer-based model (MiT-B0) benefits much more with synthetic data than CNN-based models, as shown in Figure 8 (b). When using synthetic data that is 20% of the size of the original training set, there is an enhancement in the results. Additionally, it's worth noting that the optimal amount of synthetic data required can vary based on the specific category of images. On a holistic scale, integrating 100% of the synthetic data appears to be a reasonable choice.

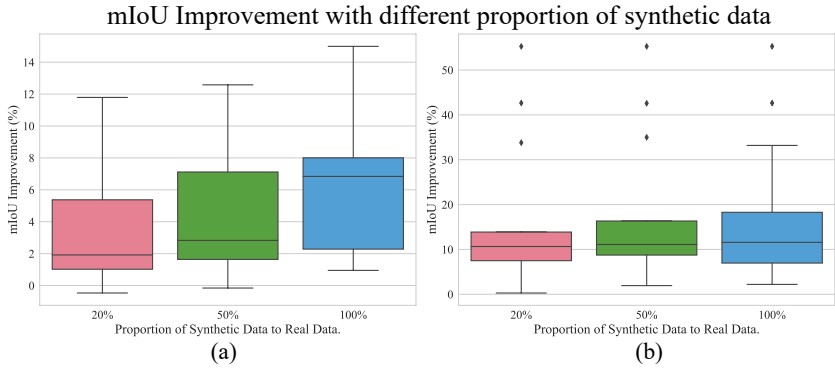

Figure 8: Improvement in mIoU with different proportions of synthetic Data. This experiment is done on Defect Spectrum (MVTec) with DeeplabV3+ and MiT-B0 shown as (a) and (b) respectively.

## 5 CONCLUSION

In conclusion, our introduction of the Defect Spectrum dataset, complemented by the Defect-Gen generator, addresses critical gaps in industrial defect inspection. By providing Semantics-abundant, precise, and large-scale annotations, we anticipate our contributions will foster advancements in defect inspection methodologies. The potential integration of Vision Language Models, the practical value of labeling assistant Defect-Click, coupled with the Defect-Gen's capability to mitigate data scarcity, sets the stage for more robust defect inspection systems in the future. We hope that our work serves as a valuable stepping stone for future researches, contributing to ongoing efforts to enhance quality assurance in industrial manufacturing.

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

# APPENDIX

In this supplementary, we extended our experiment to incorporate more annotation comparisons with existing datasets in Sec. G. The detailed generation settings and more quantitative analysis are discussed in Sec. H. We also include more visual cases in Sec. I to demonstrate the capacity of our framework to maintain both fidelity and diversity.

## A    EXTENDED SYNTHETIC DATA EXPERIMENTS

We extended our experiments to further demonstrate the effectiveness of the usage of the synthetic data by incorporating 3 more baselines: DeepLabv3+, Mask2Former, and Mit-B0. Results in Table 5 show a large performance increase in both MVTec and Cotton datasets, the increase is comparatively smaller in the VISION dataset, however, such increase is demonstrated in each of the sub-classes.

Table 5: Performance (mIoU) comparison between models trained with and without synthetic data. The bolded text indicates results with synthetic data.

|  | MVTec | VISION | Cotton |
|---|---|---|---|
| DeepLabV3+ | 51.58/**55.55** | 52.33/**53.46** | 48.73/**58.58** |
| Mask2Former | 45.70/**50.16** | 54.12/**55.47** | 64.09/**65.39** |
| MiT-B0 | 46.45/**56.21** | 49.62/**50.75** | 50.52/**55.86** |

## B    INCREASING RATIO OF SYNTHETIC DATA

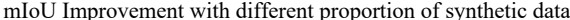

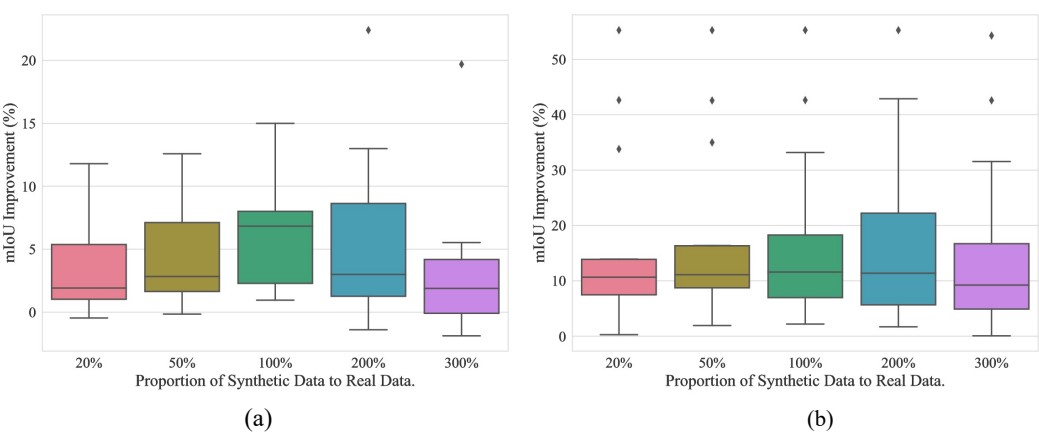

Figure 9: Improvement in mIoU with different proportions of synthetic Data. This experiment is done on Defect Spectrum (MVTec) with DeeplabV3+ and MiT-B0 shown as (a) and (b) respectively.

We further increase the synthetic data ratio to test the impact it has on model performance. Figure 9 shows the performance improvement over different quantities of synthetic data using DeepLabV3+ and Mit-B0. When using synthetic data that is 200% of the size of the original training set, there is an enhancement in the performance, but results in greater variance. Additionally, the performance starts to decrease after reaching the 300%.

## C    SAMPLING STRATEGY

We employ a new sampling strategy that uniformly up-scales all training sets to 150 images. So those with limited data gain more synthetic data and vice versa. As shown in Table 6, with this sampling strategy, MiT-B0 models show a great leap in performance, from 46.45 to 56.21, and it even surpasses the original SOTA DeepLabV3+.

Table 6: Uniform Upsampling strategy

|  | DeepLabV3+ | MiT-B0 |
| --- | --- | --- |
| Baseline | 51.58 | 46.45 |
| With Synthetic | **54.87** | **56.21** |

## D COMPARISON BETWEEN REFINED AND BASELINE DATASET

As shown in Figure 10, we compared the segmentation model trained on the baseline dataset and our refined dataset. The results with our annotation demonstrate a better granularity while having rich semantics (different defective classes).

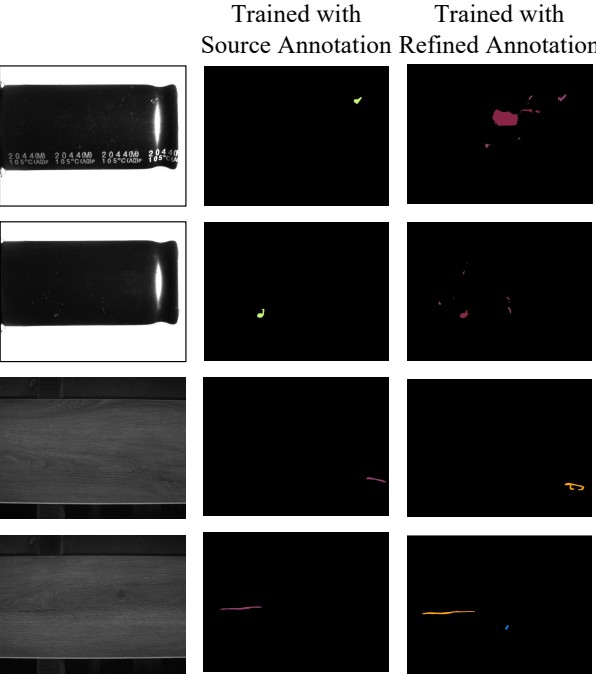

Figure 10: Qualitative comparison between the segmentation model trained on our refined dataset and the base dataset. We show our method can exhibit diversity while maintaining high quality. **Best viewed in color.**

# E  COMPARISON BETWEEN UNSUPERVISED AD AND OURS

As shown in Figure 11, we have made the comparison to the unsupervised AD baseline PaDiM. The results demonstrate using our refined dataset 1) a more granular segmentation mask; 2) a more comprehensive detection (including those that remain unlabeled in the baseline dataset); 3) captures a wide range of defective types with better granularity while having rich semantics (different defective classes).

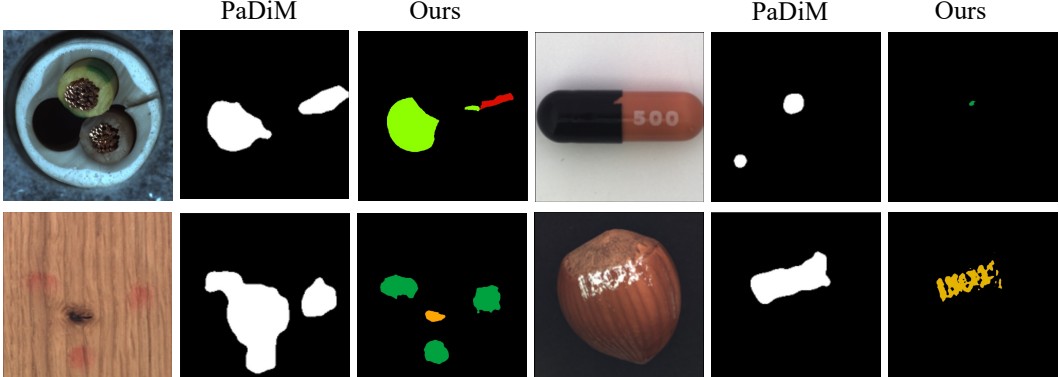

Figure 11: Qualitative comparison between the unsupervised AD baseline PaDiM and ours. Ours is trained on refined MVTec using MiT-B0. The segmentation results using our refined dataset highlighted different defective types with superior granularity. **Best viewed in color.**

# F  INCORPORATING NORMAL DATA

We conducted additional experiments on refined-MVTec to analyze the impact of integrating normal data. Our findings indicate that adding normal data indeed improves the mIoU, addressing the issue of over-penalizing non-defective areas.

Table 7: Combining different percentage of normal(defect-free) data. The source indicates the refined MVTec training set without any normal data.

|  | Source | +20% normal | +100% normal | +200% normal | +300% normal |
|---|---|---|---|---|---|
| **Mean** | 51.58 | **53.87** | 53.06 | 53.38 | 53.04 |

# G  ANNOTATION COMPARISION

In this section, we present a visual comparison between ours (the last row) and the original datasets' annotation. Figure 12, 13, 14 shows the comparison of the MVTec dataset, we re-classify the defects based on their type and enabled more semantic abundance. As for Figure 15 of VISION dataset, we refined the original annotation for more granularity. The original DAGM and Cotton datasets contained no pixel-level annotation, thus we provide our annotation as shown in Figure 16, 17.

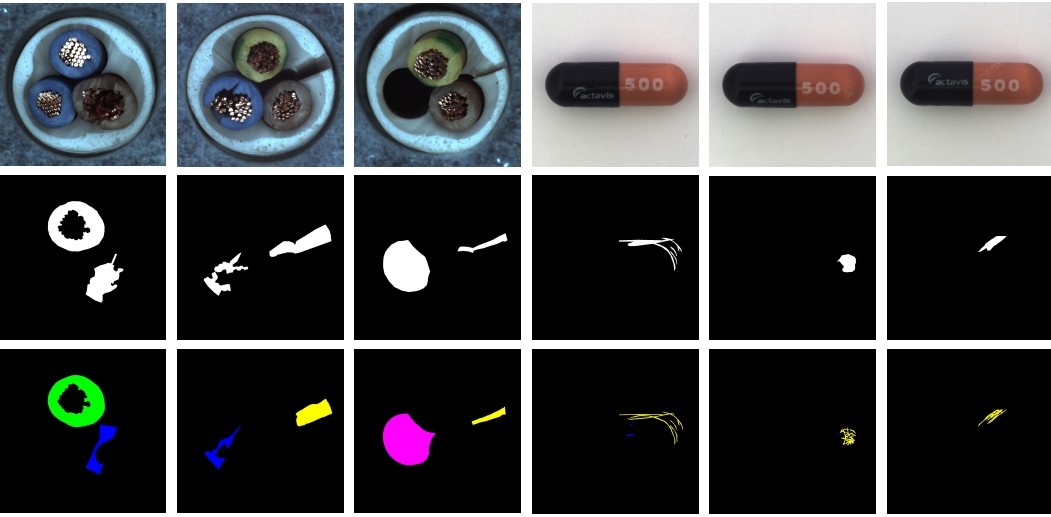

Figure 12: The annotation comparison of the "cable" and "capsule" class in MVTec dataset. The first row shows the defect image. Row 2 and 3 shows the original annotation and our improved annotation. **Best viewed in color.**

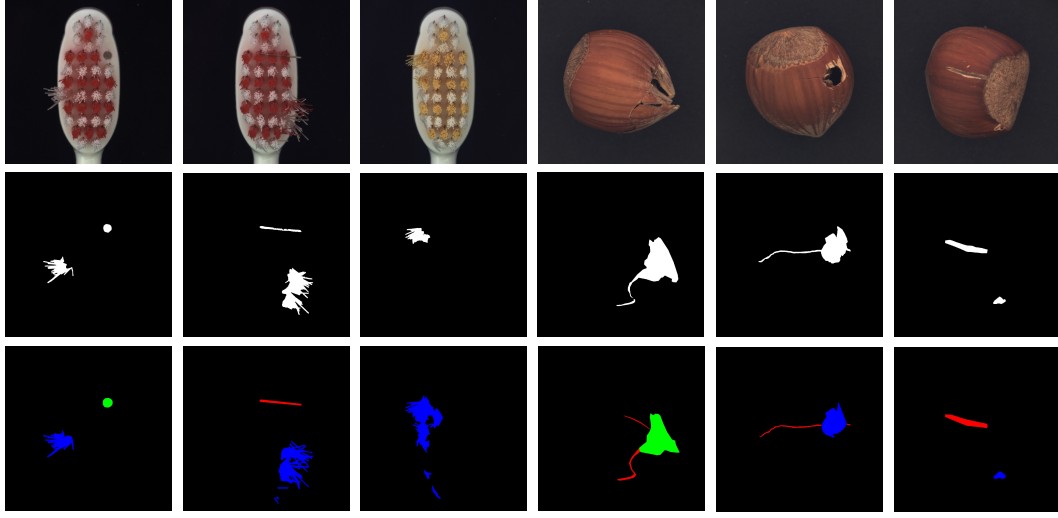

Figure 13: The annotation comparison of the "toothbrush" and "hazelnut" class in MVTec dataset. The first row shows the defect image. Row 2 and 3 shows the original annotation and our improved annotation. **Best viewed in color.**

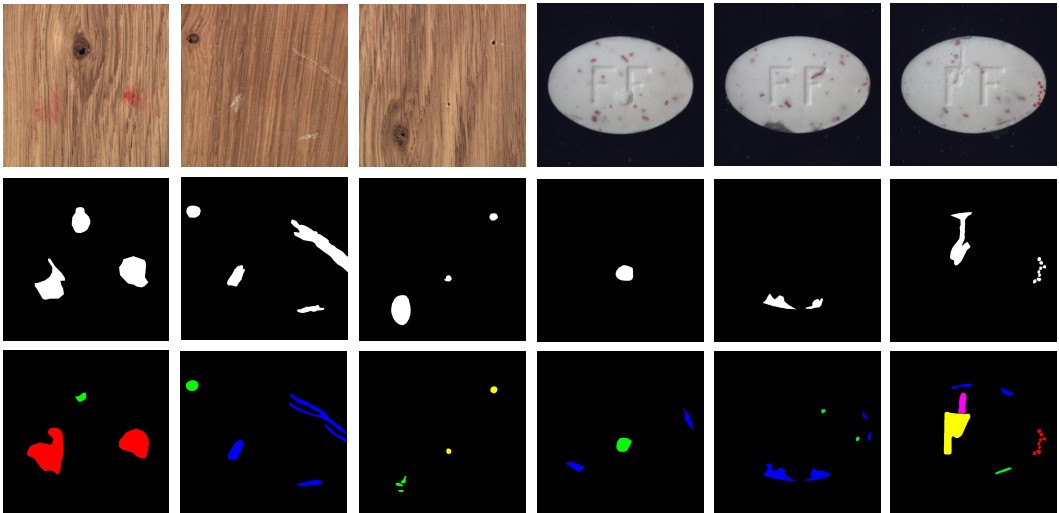

Figure 14: The annotation comparison of the "wood" and "pill" class in MVTec dataset. The first row shows the defect image. Row 2 and 3 shows the original annotation and our improved annotation. **Best viewed in color.**

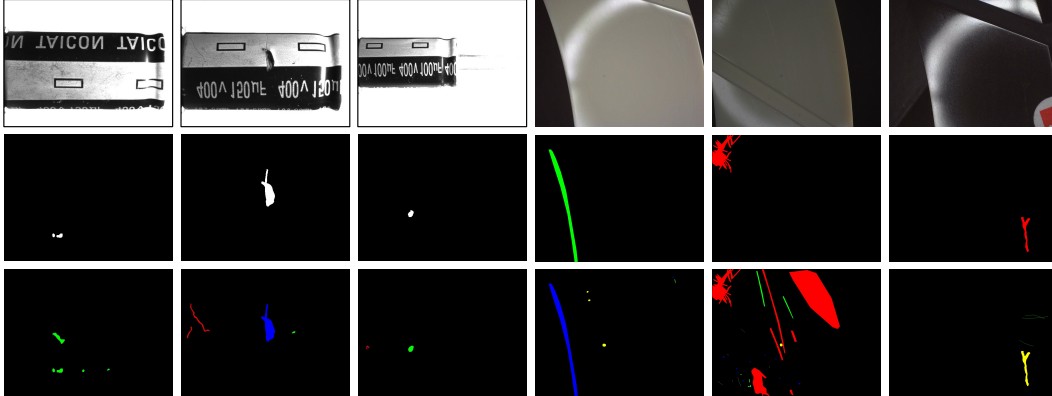

Figure 15: The annotation comparison of the "capacitor" and "ring" class in VISION dataset. The first row shows the defect image. Row 2 and 3 shows the original annotation and our improved annotation. **Best viewed in color.**

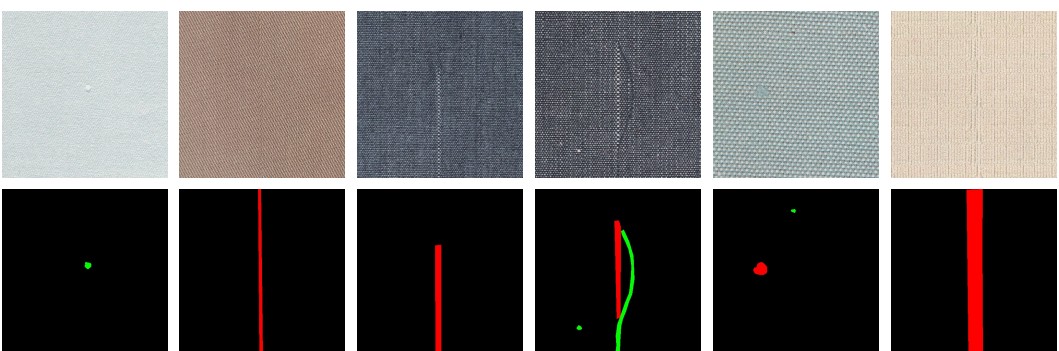

Figure 16: The annotation comparison of the "cotton fabric" class in the COTTON dataset. The first row shows the defect image. Row 2 shows our improved annotation. **Best viewed in color.**

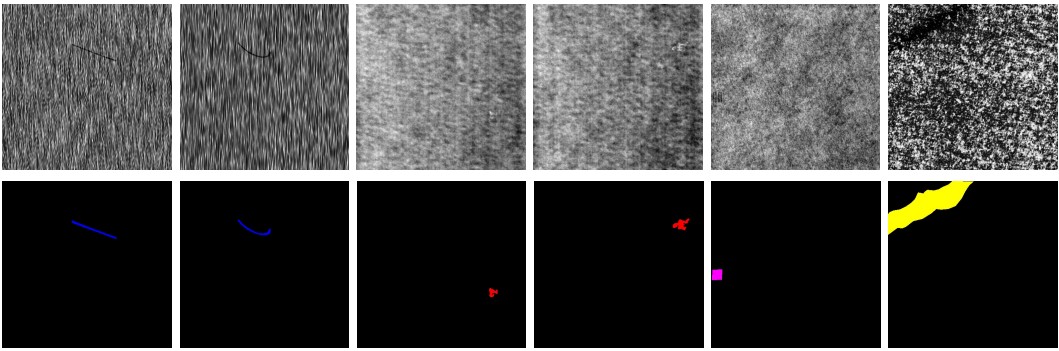

Figure 17: The annotation comparison of the "texture surface" in DAGM dataset. The first row shows the defect image. Row 2 shows our improved annotation. **Best viewed in color.**

## H DEFECT GENERATION

### H.0.1 IMPLEMENTATION DETAILS

In this section, we will first elaborate on the architecture of Defect-Gen. Then we will go over the dataset and training settings of our model. Lastly, we quantitatively compared it with other methods to demonstrate the superiority of our method.

**Experimental Settings**  Since there was no train-test split in MVTec AD dataset, to train both large and small diffusion models, we employed 5 images for each defective type per object, which is the same as our segmentation training setting. For VISION, DAGM2007, and Cotton-Fabric, we use the pre-split training set. Table 8 to 11 show the architectures of the large and small-receptive-field models. The training of diffusion models is performed on four 3090 GPUs, with a batch size of 2, a learning rate of $1e-4$, and a training iteration number of 150,000. We utilize the Adam optimizer with a weight decay of $2e-3$.

Table 8: Upsampling Block

| Layer Type | Input size | Output size | Norm | Activation |
|---|---|---|---|---|
| ResBlock $\times 2$ | $H \times W \times C$ | $H \times W \times C$ | GN | SiLU |
| Interpolation | $H \times W \times C$ | $2H \times 2W \times \frac{C}{2}$ | None | None |

Table 9: Downsampling Block

| Layer Type | Input size | Output size | Norm | Activation |
|---|---|---|---|---|
| ResBlock $\times 2$ | $H \times W \times C$ | $H \times W \times C$ | GN | SiLU |
| Avg_pool $2 \times 2$ | $H \times W \times C$ | $\frac{H}{2} \times \frac{W}{2} \times 2C$ | None | None |

**Parameter analysis**  As we discuss in Sec.3.4.2, our model has two key hyperparameters: the switch timestep $u$ and the receptive field of the small model. Both of them can control the trade-off between fidelity and diversity. We use FID to measure the generation fidelity. Since there is

Table 10: Architecture for Large receptive fields model.

| Layer Type | Resolution | # of Channels | Norm | Activation |
|---|---|---|---|---|
| InConv | 256 | 4 | GN | SiLU |
| DownSampleBlock | 256 | 192 | None | None |
| DownSampleBlock | 128 | 384 | None | None |
| DownSampleBlock | 64 | 768 | None | None |
| DownSampleBlock | 16 | 1536 | None | None |
| UpSampleBlock | 16 | 768 | None | None |
| UpSampleBlock | 64 | 384 | None | None |
| UpSampleBlock | 128 | 192 | None | None |
| UpSampleBlock | 256 | 96 | None | None |
| OutConv | 256 | 4 | GN | SiLU |

Table 11: Architecture for Small receptive fields model.

| Layer Type | Resolution | # of Channels | Norm | Activation |
|---|---|---|---|---|
| InConv | 256 | 4 | GN | SiLU |
| DownSampleBlock | 256 | 192 | None | None |
| DownSampleBlock | 128 | 384 | None | None |
| UpSampleBlock | 128 | 192 | None | None |
| UpSampleBlock | 256 | 96 | None | None |
| OutConv | 256 | 4 | GN | SiLU |

no existing metric to effectively measure the generation diversity, we used LPIPS score to indicate such. A higher LPIPS score with a similar FID score demonstrated a higher diversity in the dataset. Table 12 shows the FID and LPIPS for different $u$ and receptive fields. As shown, when $u$ increases, fidelity increases while diversity decreases. Similarly, when the receptive field switches from small to large, the same trend occurs. Empirically, we use $u$=50 and the medium receptive field to achieve a good trade-off between FID and LPIPS.

Table 12: The table shows the trade-off between diversity and image quality of the capsule class. The column represents 3 different receptive field sizes, large, medium, and small, and the respective down-sampling blocks are 6, 3, 2. The row represents the timesteps($v$) used for the small receptive field model.

| | u | 25 | 50 | 75 | 100 | 400 | 700 |
|---|---|---|---|---|---|---|---|
| Small | FID ↓ | 115.2754 | 93.2839 | 80.8040 | 79.6411 | 82.5127 | 78.4115 |
| | LPIPS ↑ | 0.3981 | 0.3666 | 0.3537 | 0.3523 | 0.3467 | 0.3460 |
| Medium | FID ↓ | 69.9419 | 57.5374 | 57.3961 | 57.8977 | 57.426 | 57.006 |
| | LPIPS ↑ | 0.3473 | 0.3458 | 0.3450 | 0.3417 | 0.3392 | 0.3381 |
| Large | FID ↓ | 59.085 | 56.6246 | 56.7247 | 56.2493 | 55.7226 | 54.0529 |
| | LPIPS ↑ | 0.2914 | 0.2870 | 0.2866 | 0.2853 | 0.2832 | 0.2814 |

### H.0.2 QUANTITATIVE EVALUATION

We have compared the segmentation performance boost across different methods on the original MVTec dataset. GAN-based methods were excluded since they hardly generate realistic images, further disrupting the original data distribution. Results for defect segmentation are shown in Table. 13. The first column shows the defect segmentation mIoU score with only the original training data. The rest of each column presents defect segmentation performance with original training data pairs and the augmented pairs generated by different synthesis methods. SinDiffusion dropped the mIoU score, due to the incorrectly structured output images and mislabeled masks. However, it can slightly improve the segmentation performance for certain classes like "Carpet", "Grid", "Leather", "Tile" and "Wood". Since those classes do not contain any industrial parts and thus do not require any global structure information during synthesizing. DDPM-generated samples can boost the performance score, however, due to the lack of diversity during generation, the increase in performance is limited.

Table 13: Quantitative comparison on segmentation performance between sinDiffusion, DDPM, and our method. To demonstrate the effectiveness of our method on other dataset besides Defect Spectrum, the comparison was made on the original MVTec dataset

|  | w/o any AUG | sinDiffusion | DDPM | Ours |
|---|---|---|---|---|
| capsule | 75.47 | 76.25 | 79.21 | 82.20 |
| bottle | 67.54 | 70.52 | 67.32 | 73.75 |
| carpet | 67.33 | 72.89 | 68.94 | 74.27 |
| screw | 53.12 | 49.66 | 60.12 | 58.78 |
| grid | 59.68 | 61.59 | 60.68 | 62.14 |
| cable | 46.28 | 41.75 | 48.28 | 49.14 |
| hazelnut | 69.25 | 65.65 | 69.25 | 71.46 |
| leather | 66.39 | 66.91 | 66.39 | 66.80 |
| metal_nut | 69.56 | 63.5 | 68.57 | 74.4 |
| pill | 69.71 | 66.75 | 70.14 | 73.19 |
| tile | 70.33 | 72.43 | 71.23 | 73.58 |
| toothbrush | 68.26 | 64.26 | 68.09 | 70.14 |
| transistor | 44.31 | 47.16 | 44.37 | 47.47 |
| wood | 65.33 | 70.25 | 64.93 | 68.55 |
| zipper | 67.62 | 63.12 | 68.61 | 70.48 |
| **mean** | 64.01 | 63.51 | 65.07 | 67.76 |

### H.0.3 DEMONSTRATION OF PATCH-LEVEL MODELING

Figure 18 demonstrates the effectiveness of this strategy. Overall, the generated sample is different from the training samples, while at the patch level, we can find some connections in between.

## I VISUAL GENERATION RESULTS

We have included more defect generation results along with their masks as shown in Figure 19 to 24 below.

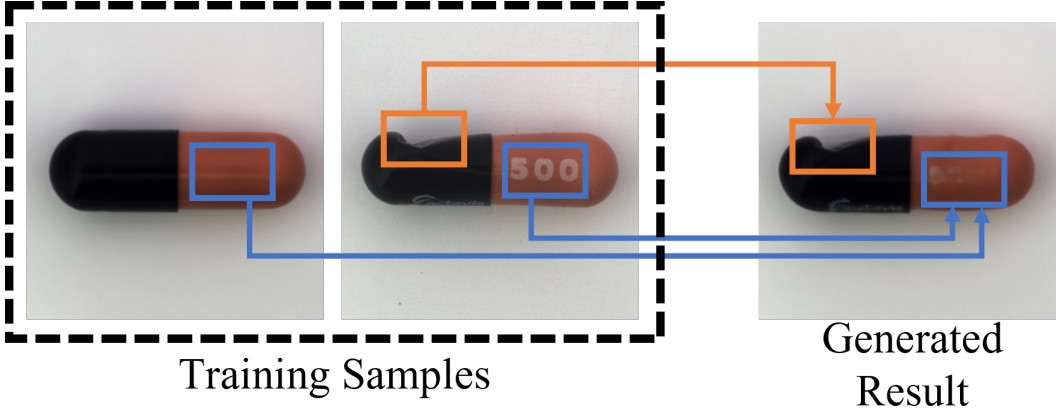

Figure 18: The property of patch-level modeling. The right image is generated from the small-receptive-field model, and the two left images are the two most similar images from the training set.

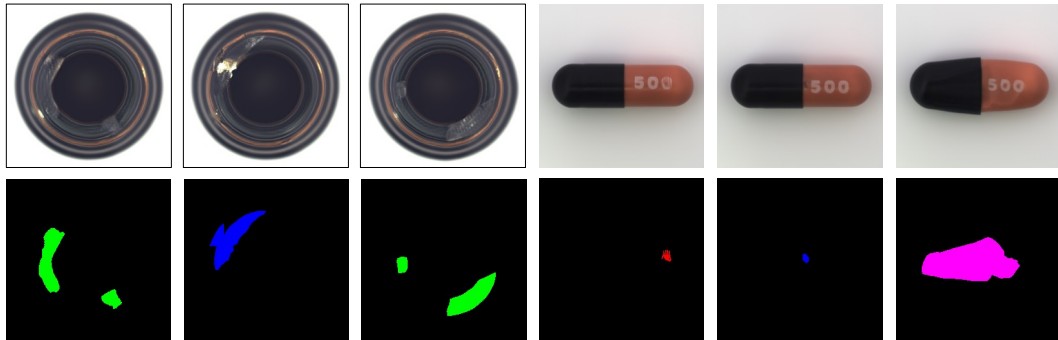

Figure 19: The generated images and masks of the "bottle" and "capsule" class. **Best viewed in color.**

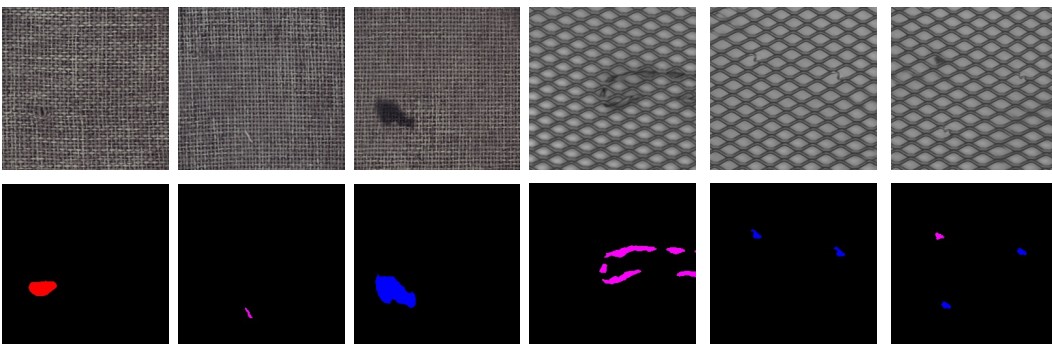

Figure 20: The generated images and masks of the "carpet" and "grid" class. **Best viewed in color.**

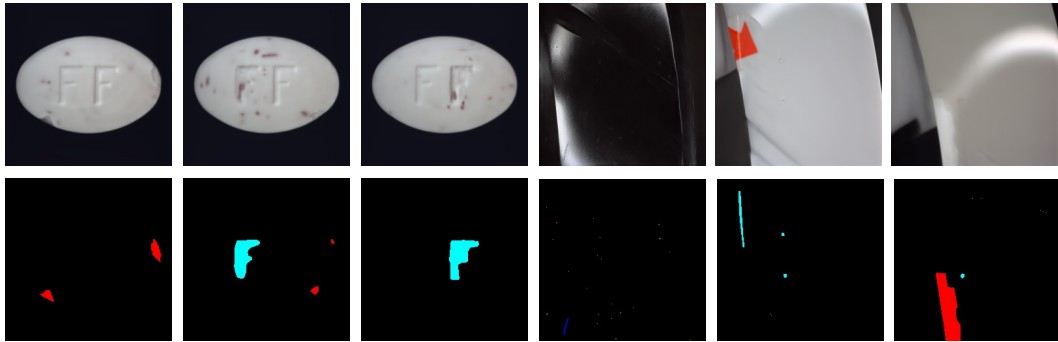

Figure 21: The generated images and masks of the "pill" and "ring" class. **Best viewed in color.**

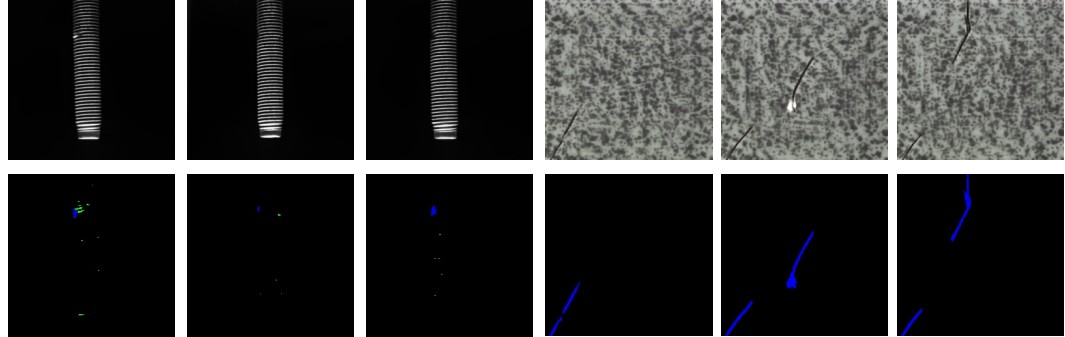

Figure 22: The generated images and masks of the "screw" and "tile" class. **Best viewed in color.**

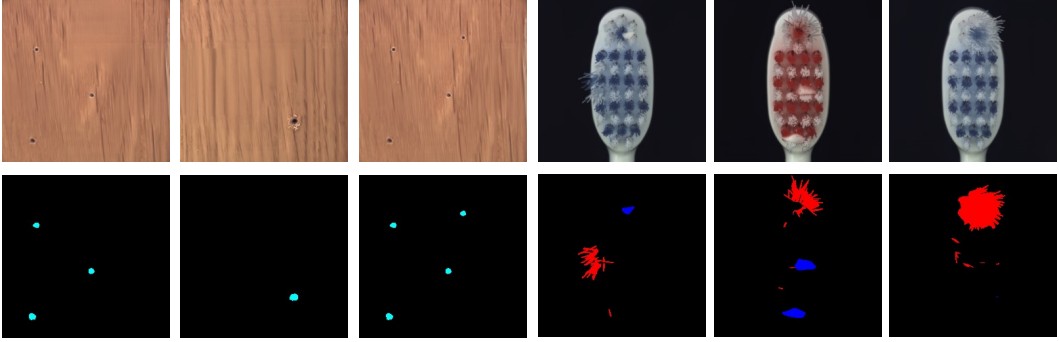

Figure 23: The generated images and masks of the "wood" and "toothbrush" class. **Best viewed in color.**

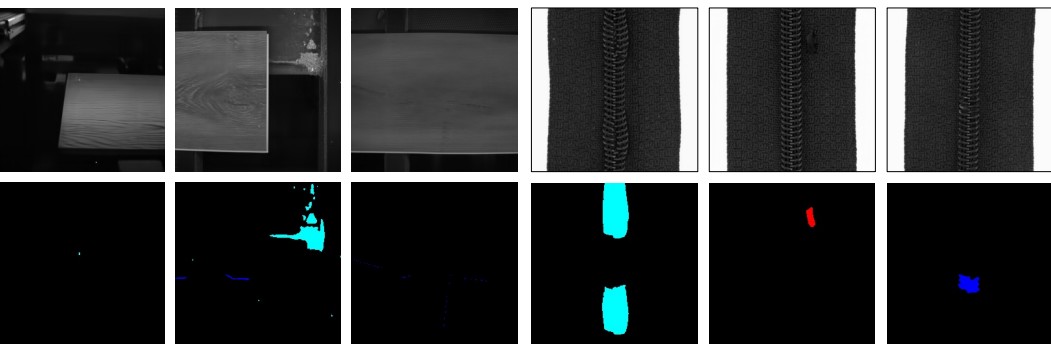

Figure 24: The generated images and masks of the "wood-surface" and "zipper" class. **Best viewed in color.**

