# OpenReview forum: "Defect Spectrum: A Granular Look of Large-Scale Defect Datasets with Rich Semantics"
_ICLR.cc/2024/Conference — Submitted to ICLR 2024_

### Official Review · Reviewer_EDJD · 2023-10-28

**Soundness:** 2 fair
**Presentation:** 3 good
**Contribution:** 3 good
**Rating:** 5
**Confidence:** 4

**Summary:**

The paper integrated previous public datasets and designed an annotation tool for fast annotation, thereby adding rich semantic annotations to the dataset. To further increase the diversity of defective images, the paper proposed a two-stage Diffusion Model to generate additional abnormal data. In addition, the paper conducted extensive experiments on segmentation methods based on CNN and ViT, establishing a reliable benchmark. Finally, the paper validated that the proposed data augmentation method improves the performance of segmentation methods.

**Strengths:**

+ The paper first proposes to add rich semantic information to pixel-level annotation. As far as I know, this is the first work in defect detection that performs semantic annotation at the pixel level.
+ To achieve the goal of adding semantic-level pixel annotation, the paper designed a tool for defect annotation using SAM and Focal-Click.
+ To enrich defective images, the paper designed a two-stage Defect Synthesis Diffusion Model.
+ The paper conducted segmentation experiments on various segmentation models and established a reliable benchmark.

**Weaknesses:**

+ The paper focused on semantic annotation of defects, but completely ignored the utilization of normal data. As a result, the dataset for defect detection is not much different from semantic segmentation datasets, which leads to the task of defect detection becoming more similar to semantic segmentation of natural images. This needs to be considered.
+ Although the paper added rich semantic information to the dataset, its detection performance was not very good. Existing unsupervised defect detection methods have already surpassed an AUPR score of 0.7 at the pixel level, which means that existing defect detection methods can achieve an IoU of over 70 without using defect data. Some methods that use a large amount of normal data and a small amount of abnormal data have even higher performance.

**Questions:**

+ I think the motivation behind the paper is very valuable, but perhaps the authors need to consider the differences from existing unsupervised anomaly detection methods.
+ Although there are few methods that use abnormal data for detection, some methods still use a large amount of normal data and a small amount of abnormal data. Their main weakness is the inability to determine the type of defect, which may be a problem that the paper can solve. However, the paper may need to do some additional work beyond the data to design a solution for this problem.

---

> ### Author Response · Authors · 2023-11-18
>
> Thank you for your thorough and insightful feedback on our paper. Your expertise in unsupervised anomaly detection is evident, and we appreciate the depth of your review. Firstly, we’d like to revisit our work. This paper primarily aims to pose a new question in defect detection research, rather than provide a definitive solution. We observed that current benchmarks in defect inspection research are approaching performance saturation. However, in real-world scenarios, defect inspection remains a challenging problem. This gap between academic research and industry applications prompted us to propose a new, more challenging benchmark. In this paper, we invite the research community to explore algorithms that can effectively address this benchmark.
> As for your concerns. we fully understand and would like to address them as follows:
>
> **Inclusion of Anomaly Detection (AD) Methods**: In our benchmark, we focused on evaluating semantic segmentation methods, intentionally excluding unsupervised AD methods. The reason is that current AD methods are not equipped to provide masks for different defect types, which is crucial in our approach. We compute the mean Intersection over Union (mIoU) by averaging across different defect types, a metric where AD methods fall short as they cannot differentiate between these types.
> Regarding the concern about 'poor detection performance,' our use of mIoU for multi-class segmentation sets a higher standard than the binary classification task's Average Precision at a Recall (AUPR). Although our mIoU values are lower than AUPR scores reported in AD papers, they reflect a more robust defect inspection capability, as demonstrated in **Appendix Section. E** of our updated paper. Additionally, AD methods struggle with unaligned images, such as those in the VISION dataset, where variance is high.
>
> **Utilizing Normal Data**: We agree with your observation regarding the utilization of normal data. Normal data are indeed part of our defect spectrum dataset, initially included in our four subsets without requiring additional annotation. To address your concern about performance benchmarking, we have decided to incorporate normal data into our training set and re-evaluate our candidate methods. Due to time constraints in the rebuttal period, we have re-conducted a portion of the experiments from Table 3. Preliminary results shown in the table below (also see **Appendix Section. F** of our updated paper) demonstrate improved performance with the inclusion of a small amount of normal data by addressing the issue
> of over-penalizing non-defective areas. We plan to revise Table 3 accordingly, incorporating these findings.
>
> Combining different percentages of normal(defect-free) data. The source indicates the re-fined MVTec training set without any normal data
> |      | Source | +20% Normal       | +100% Normal  | +200% Normal  | +300% Normal  |
> |--------|--------|-------------------|---------------|---------------|---------------|
> | **Mean** | 51.58  | **53.87**        | 53.06         | 53.38         | 53.04         |
>
> **Differences from Semantic Segmentation**: While our task may appear similar to semantic segmentation, there are distinct challenges in handling industrial data. These include issues such as few-shot learning and the small area problem, which differentiate our task from standard semantic segmentation in natural images.
>
> **Future Directions and Community Involvement**: Our work aims to define a more complex defect inspection task in a practical manner and present augmented datasets for exploration. We leave the design of specific methods to the broader research community. Our paper's goal is to establish a foundation for researchers to adapt and leverage state-of-the-art methods within our proposed benchmark.
>
> Finally, I sincerely appreciate the valuable insights provided by the reviewers and warmly welcome any further criticisms or suggestions. Your timely feedback is crucial for the enhancement of this paper. I look forward to your valuable comments.

---

> ### Author Response · Authors · 2023-11-21
>
> Dear reviewer EDJD,
>
> As the window for reviewer-author interaction is closing soon, on November 21st, I wanted to extend my sincerest gratitude for the invaluable time and effort you have dedicated to reviewing our work. To ensure that we have met your expectations, may I kindly ask if you find our responses satisfactory and if there are any remaining issues that need further clarification or improvement?

---

> ### Comment · Reviewer_EDJD · 2023-11-21
>
> Thanks to the author for his detailed answer. Although there is a large gap between this work and existing research, I believe it is a valuable attempt. The experimental results that the author added using more normal samples are in line with expectations, but what I want more is some insight into how to better utilize normal samples. Taken together, I think the paper is a meaningful attempt. However, it's might suitable for some dataset track or workshop.

---

> ### Author Response · Authors · 2023-11-22
>
> Dear reviewer EDJD,
>
> Thank you for your invaluable comment. I noticed that you change the rating from 6 to 5 and added a comment of “However, it's might suitable for some dataset track or workshop.”
>
> I would like to clarify a key point regarding the submission category of our paper. We have submitted our paper under the **"Primary Area: datasets and benchmarks"** track. This submission choice aligns with the guidelines of ICLR, which, to our understanding, accepts papers focused on datasets.
>
> Given this alignment with the submission track requirements, I kindly request you to reconsider the rating of our paper. We believe that our submission is well-placed in the datasets and benchmarks track, and we hope this clarification might positively influence your assessment.

---

> > ### Comment · Reviewer_EDJD · 2023-11-23
> >
> > Thank you for your response. I changed my score primarily based on the opinion of **Reviewer VeNd**. The paper focuses mainly on datasets and benchmarks, but additional annotation work was done on other people's datasets. Besides, these captions and semantic annotations do not demonstrate an advantage over unlabeled data. The proposed annotation tool and generation method in the paper will contribute to the development of the field, but as a new dataset, the paper cannot complete the closed loop of problem formulation and solution.

---

> > > ### Author Response · Authors · 2023-11-23
> > >
> > > Thanks for your feedback. For your new concerns, we would like to address them one by one.
> > >
> > > **The closed loop of problem formulation and solution.**
> > >
> > > Firstly, we would like to revisit the loop of problem formulation and solution.
> > >
> > > 1) We discover that there is a gap between the academy and the industry. In industry, identifying defect class and size is very crucial. Analysis can be found in paragraph 2 of our introduction section.
> > >
> > > 2) However, current methodologies and datasets are inadequate for bridging this gap. Specifically, existing Anomaly Detection (AD) methods fall short in differentiating among defect classes. Semantic segmentation emerged as a promising solution, yet it too fails to deliver precise results due to the low quality of annotations available.
> > >
> > > 3) Therefore, we propose our refined datasets to fill this gap in a data-centric way. We provide annotations with rich semantics and introduce two innovative tools: 'Defect-Click', designed to assist annotators, and 'Defect-Gen', which efficiently generates additional image-mask pairs.
> > >
> > > 4) The application of our enhanced datasets significantly improves the ability of segmentation models to differentiate between various defect classes and provide accurate results. This advancement effectively bridges the previously identified gap, as evidenced by the benchmarks in Table 3 and the comparative analysis in Appendix D.
> > >
> > > In summary, our work tackles this industry problem from a data perspective. All our proposed methods serve for an effective dataset. The above step-by-step analysis forms the closed loop of our work.
> > >
> > > **The advantages of our refined dataset**
> > >
> > > It is hard to quantitatively compare the baseline datasets with our refined dataset. Our refined dataset has more classes (including training and validation sets). It is not practical to compare a model trained on the baseline dataset with ours. Given MVTec toothbrush class for example, the baseline training/validation set only has one defective class, but for our refined toothbrush class, there are three defective classes. Thus, it would be not reasonable to have a model trained on the baseline toothbrush dataset and run validation on our refined dataset, and vice versa.
> > >
> > > Therefore, we tried our best to demonstrate the superiority of our refined dataset in Appendix D, E, G. In Appendix. D, we made a comparison between the model trained on the baseline and refined datasets respectively, where some small defective areas were being ignored in the baseline model but were successfully being identified using our model. Appendix E illustrates the comparison between the unsupervised baseline model, PaDiM, and our refined model, demonstrating that the latter not only successfully identify defect classes but also yields more precise detection results. Finally, Appendix G showcases a comparison between baseline annotations and our refined annotations. It underscores that our refined dataset meticulously annotates smaller defects, which, although minimal in size, can have significant impacts in the manufacturing process and were previously neglected in the baseline dataset.
> > >
> > > **Clarification of basing on other datasets**
> > >
> > > In our definition of the problem, the defect inspection mask is paramount as it plays the most critical role in bridging the gap between academia and industry. While we utilize images from existing datasets, it is important to note that all annotations are entirely generated by our team.
> > >
> > > **Concerns raised by Reviewer VeNd**
> > >
> > > We believe we have addressed all the concerns from **Reviewer VeNd.** Please see our comments under his/her review for details.

---

### Official Review · Reviewer_VeNd · 2023-10-30

**Soundness:** 2 fair
**Presentation:** 1 poor
**Contribution:** 1 poor
**Rating:** 1
**Confidence:** 4

**Summary:**

This paper refines the existing defect inspection datasets by introducing more detailed annotations and additioinal synthetic data from the diffusion generator. The experimental results demonstrate partial effectiveness of the refined datasets.

**Strengths:**

+ This paper provides more detailed annotations for the existing defect inspection datasets, which can partially be benefit to the development of defect inspection.

**Weaknesses:**

+ This paper is more suitable for a demos track, which only displays its rough progression of dataset refinement, but lacks technique detail presentation. The important details about auxiliary annotation tool and diffusion generator for defect image generation are not clearly clarified. If this refined dataset and annotations are private (not publicly available), the contributions to the development of defect inspection are very limited.

+ The comparative experiments between the refined dataset and the baseline dataset are missing in Tab.3. The results in Tab.3 can not illustrate the advantages of the refined dataset.

+ The results of VISION dataset in Tab.4 is somewhat trival. The authors should analyze these results. The improvements of refined dataset should be clearly clarified, such as how many synthetic data are supplemented and the performance improvement relative to the increment of dataset size.

**Questions:**

See in Weaknesses.

---

> ### Author Response · Authors · 2023-11-18
>
> Thank you for your insightful observations. We address each of your concerns as follows:
>
> **Open Source Issue**: We completely understand your concern regarding the open-source of our dataset. We have now made these resources publicly through this anonymous link: https://defect-spectrum-authors.github.io/defect-spectrum/, ensuring that our contributions to the field of defect inspection are substantial and accessible to the research community.
>
> **Comparison between Refined and Baseline Dataset**: To address your concern about the comparative experiments. Since both the training set and testing set were being refined, it is hard to compare the baseline-trained model quantitatively using the refined testing set or the source testing set. Thus, we qualitatively compared the segmentation model trained on the baseline dataset and our refined dataset, as shown in the **Appendix section. D** of our updated paper. The results with our annotation demonstrate 1) a more granular segmentation mask; 2) a more comprehensive detection (including those that remain unlabeled in the baseline dataset); 3) captures a wide range of defective types with better granularity while having rich semantics (different defective classes).
>
> **Clarification on Table 4 and Synthetic Data Utilization**: We appreciate your comments on Table 4 and the need for clarification regarding synthetic data. Regarding the results in Table 4, we would like to clarify that we have selected the best baseline to perform our performance boost experiment. We think although the improvement is comparatively smaller than the increase in the MVTec dataset, it still demonstrates the effectiveness of the synthetic data. We also extended the experiment to compare using three other baselines: M2F, MiT-B0, and DeepLabV3+, the experiments can be found in the table below (also see **Appendix Section. A** in the updated paper).
>
> Performance (mIoU) comparison between models trained with and without synthetic data. The bolded text indicates results with synthetic data.
> | Model        | MVTec                     | VISION                    | Cotton                     |
> |--------------|---------------------------|---------------------------|----------------------------|
> | DeepLabV3+   | 51.58/**55.55**           | 52.33/**53.46**           | 48.73/**58.58**            |
> | Mask2Former  | 45.70/**50.16**           | 54.12/**55.47**           | 64.09/**65.39**            |
> | MiT-B0       | 46.45/**56.21**           | 49.62/**50.75**           | 50.52/**55.86**            |
>
> **Generation Data Ratio Experiment**: Further, as shown in Figure. 8, our experiments include a detailed analysis of the generation data ratio, where we examine the impact of varying proportions of synthetic data on model performance. This provides a comprehensive view of how synthetic data supplementation affects defect detection capabilities. We are sorry for the confusion it brought and will adjust it by extending the experiment as we mentioned in the response to reviewer 2(qXne), the results for the extended synthetic data ratio experiment can be found in **Appendix Section. B** of our updated paper.
>
> Finally, I sincerely appreciate the valuable insights provided by the reviewers and warmly welcome any further criticisms or suggestions. Your timely feedback is crucial for the enhancement of this paper. I look forward to your valuable comments.

---

> ### Author Response · Authors · 2023-11-21
>
> Dear reviewer VeNd,
>
> As the window for reviewer-author interaction is closing soon, on November 21st, I wanted to extend my sincerest gratitude for the invaluable time and effort you have dedicated to reviewing our work. To ensure that we have met your expectations, may I kindly ask if you find our responses satisfactory and if there are any remaining issues that need further clarification or improvement?

---

> ### Comment · Reviewer_VeNd · 2023-11-21
>
> Thanks for your response. My concens are not well addressed.
> + The publication of captions and synthetic data. The authors only publish the refined annotation about 4 public datasets. The most important captions and synthetic data are totally unavailable. The method for data generation is the most important part for this paper. However, the solution and synthetic data are not well clarified.
> + The source code only provides the diffusion generator for configuration file of cotton fabric dataset. The configurations for other datasets are unavailable.
> + The important details about auxiliary annotation tool and diffusion generator for defect image generation are not clearly clarified.
> + The visualization results cannot effectively support the superority of the proposed dataset. I would like to see more quantitative results.
> + I still think the results of VISION dataset in Tab.4 is somewhat trival.
>
> Hence, I keep the previous score.

---

> > ### Author Response · Authors · 2023-11-22
> >
> > **The publication of captions and synthetic data.**
> >
> > We appreciate your feedback, and we've made updates to our HuggingFace repository. This includes the addition of all synthetic data for and captions for MVTec data.
> >
> > Regarding the captions for VISION and Cotton-Fabric, we are currently working on improving their quality. Rest assured that we are committed to releasing them once the refinement process is complete.
> >
> > **More configuration files**
> >
> > Thanks for pointing this out, we have updated our repository and included the configuration file for all datasets.
> >
> > **Details on defect-gen**
> >
> > As for the defect generation, we have included the technical and implementation details in **Section 3.4** as well as **the Section. H in our Appendix**. In addition to this, we have released the full implementation for our defect generator, could you provide more on what level of details you want us to elaborate?
> >
> > **Details on Defect-Click**
> >
> > Section 3.3 shows the details of Defect-Click. It a simple tools that reuse all the implementation of Focal-Click and added with our improvements. Specifically, our improvements include integrating 21 proprietary labeled datasets, introducing multi-level crop training for small defects, and incorporating edge-sensitive losses during training. The 21 proprietary labeled datasets are some defective image-mask pairs for industrial inspection. Multi-level crop training means we rescale the training samples randomly to a resolution of [512, 1024, 1536, 2048, 2560, 3072] and then crop 512x512 patches for training. Edge-sensitive losses denote the loss function in Mask2Former [1]. We use the cls_loss:mask_loss:dice_loss = 2:5:5 in practice.
> >
> > **Quantitaive evaluation of the refined dataset**
> >
> > It is hard to quantitatively compare the baseline datasets with our refine dataset. Our refined dataset has more classes (including training and validation set).  Thus, it is not practical to compare a model trained on the baseline dataset with ours. Given MVTec toothbrush class for example, the baseline training/validation set only has one defective class, but for our refined toothbrush class, there are three defective classes. Thus, it would be not reasonable to have a model trained on the baseline toothbrush dataset and run validation on our refined dataset, and vice versa. We’d like to hear some suggestions about quantitative comparison.
> >
> > **Discussion on mIoU improvement:**
> >
> > As for the performance boost regarding the usage of synthetic data on VISION dataset, we think such an increase is not trivial given other works that incorporating synthetic data for boosting segmentation performance, like FreeMask [2] and DiffuMask [3]. In FreeMask [2], the average performance increase with synthetic data is 1.9 on ADE20K and 1.28 on COCO-stuff. The average performance increase described in DiffuMask [3] is 1.5 on VOC2012, 1.4 on CityScapes. Thus, we believe we have achieved a reasonable increase in the VISION data with regard to the incorporation of the synthetic data.
> >
> > [1] "Masked-attention mask transformer for universal image segmentation." Cheng, Bowen, et al., CVPR 2022
> >
> > [2]”FreeMask: Synthetic Images with Dense Annotations Make Stronger Segmentation Models”, Yang et al, NeurIPS 2023
> >
> > [3]”DiffuMask: Synthesizing Images with Pixel-level Annotations for Semantic Segmentation Using Diffusion Models”, Wu et al, ICCV 2023

---

> > > ### Comment · Reviewer_VeNd · 2023-11-23
> > >
> > > Additionally, I suggest that the authors should change the "Large-Scale Defect Datasets" in the paper title.
> > > The size of dataset is below 10K.  The description of  "Large-Scale Defect Datasets" is overclaimed.

---

> > > > ### Author Response · Authors · 2023-11-23
> > > >
> > > > Thanks for your suggestions. We recognize that large-scale datasets for natural images typically contain a vast amount of data. However, in the field of defect inspection, gathering defective cases is exceptionally challenging, which results in most existing datasets being limited in size. To the best of our knowledge, with the exception of the unreleased VISION-V2, we hold the largest collection of annotations in the industrial domain, enriched with comprehensive semantics.
> > > >
> > > > Additionally, we kindly ask about the feedback on whether our responses have adequately addressed your concerns regarding the publication of data and configuration files, the explanations of details, the comparisons using the refined dataset, and the improvements in mIoU. Could you please clarify why these updates have not sufficiently influenced your evaluation score at all?

---

### Official Review · Reviewer_qXne · 2023-10-31

**Soundness:** 4 excellent
**Presentation:** 3 good
**Contribution:** 4 excellent
**Rating:** 8
**Confidence:** 4

**Summary:**

This paper presents the Defect Spectrum, a novel dataset that provides rich and accurate defect annotations for industrial defect inspection. The main advantages are two folds. First, it contains very accurate annotations, and very fine-grained semantics, that are critical in modern industrial manufacturing pipelines. Second, it presents a data generation solution that can generate high fidelity samples with annotations, which further increases the number of samples for training.

**Strengths:**

1. This work addresses the accuracy of masks and richness of defect annotations, which are overlooked in by previous works. The mask annotation is much more accurate than previous approaches. And to the best of my knowledge, the number of defect type is much more than previous works. This is quite important for practical industrial manufacturing.

2. The data generation approach, albeit simple, seems quite interesting. By restraining the receptive field, the proposed approach models global and patch-level distribution in a structured way, allowing for learning from few defect samples. The effectiveness is also demonstrated by experiments like Fig. 8 and Table 4.

**Weaknesses:**

1. Although the dataset contains descriptive captions, they are not used in the defect inspection baselines. It is not clear with me why these descriptive captions are essential and how they are used in practical systems? To me, it seems that captions are not as accurate as the provided semantic masks. And in what way can these captions improve the defect inspection system?

2. Figure 8 looks quite interesting, but it seems a bit incomplete. In Fig. 8, there is a clear trend that when adding more synthetic samples, the performance of both approaches is boosted. Thus, It would be more interesting if results of further increasing the ratio of synthetic data can be shown.

3. It seems the the sampling strategy for generated samples could be improved. In the paper, it seems that the quantity of generated samples is at a fixed ratio to the real samples. In this sense, if there is a lot of real data for certain object, then model would generate a lot of synthetic data, and vice versa. This is a bit counter-intuitive to me. It seems more reasonable that for those objects with limited number of real data, we should generate more synthetic data to mitigate the overfitting problem.

**Questions:**

Overall it is a good paper. Please address my concerns in the weakness section to make it stronger, e.g., make it make it clearer hof ow the captions benefit practical models, and provide more in-depth analysis to the sampling strategy of generated samples.

---

> ### Author Response · Authors · 2023-11-18
>
> Thank you for your insightful observations. We address each of your concerns as follows:
>
> **Importance of Image-Text Pair Datasets**: Image-text pair datasets are crucial for training Multimodal Large Language Models(MLLMs), which have shown their dominant capabilities in a large range of tasks. Our descriptive captions offer additional contextual information that can be leveraged by these models. We look forward to empowering MLLMs with the ability of defect inspection through this effort.
>
> **Extended Synthetic Data Ratio Experiments**: To address your concerns, we have extended our experiments to include scenarios with higher ratios of synthetic data, shown in the **Appendix Section. B**(Please refer to the updated paper). Specifically, we have now included results for systems trained with 200% and 300% synthetic data ratios. When using synthetic data that is 200% of the size of the original training set, there is an enhancement in the performance, but results in greater variance. Additionally, the performance starts to decrease after reaching 300%. This is due to the overly-used synthetic data disrupting the original data distribution.
>
> **Sampling Strategy**: As an immediate response to your suggestion, we conduct an experiment where we employ a uniform sample strategy – using a fixed number (e.g., 150) of samples in the training set for each object. So those with limited data gain more synthetic data and vice versa. As shown in the table below (also see **Appendix Section. C**), with this sampling strategy, baselines such as MiT-B0 show a great leap in performance.
>
> |       | Baseline: **DeepLabV3+** | With Synthetic | Baseline: **MiT-B0** | With Synthetic |
> |--------|--------------------------|----------------|---------------------|----------------|
> | **Mean** | 51.58             | **54.87**      | 46.45           | **56.21**      |
>
> Finally, I sincerely appreciate the valuable insights provided by the reviewers and warmly welcome any further criticisms or suggestions. Your timely feedback is crucial for the enhancement of this paper. I look forward to your valuable comments.

---

> ### Author Response · Authors · 2023-11-21
>
> Dear reviewer qXne,
>
> As the window for reviewer-author interaction is closing soon, on November 21st, I wanted to extend my sincerest gratitude for the invaluable time and effort you have dedicated to reviewing our work. To ensure that we have met your expectations, may I kindly ask if you find our responses satisfactory and if there are any remaining issues that need further clarification or improvement?

---

> > ### Comment · Reviewer_qXne · 2023-11-22
> >
> > Thank for the detailed responds. The authors have addressed my concerns and I keep my original rating.

---

### Official Review · Reviewer_J4Kk · 2023-10-31

**Soundness:** 2 fair
**Presentation:** 3 good
**Contribution:** 3 good
**Rating:** 6
**Confidence:** 4

**Summary:**

Based on four industrial defect detection benchmarks, namely MVTec, VISION V1, Cotton-Fabric, and DAGM2007, this paper creates a new hybrid dataset by refining the existing annotations and differentiating multiple defect types within each image. Additionally, this paper propose Defect-Gen, a two-stage diffusion-based generator meant to generate a diverse set of defective synthetic images when dealing with limited datasets. The authors conduct experiments to evaluate the diversity of the proposed dataset, as well as the quality of the synthetic data and its potential to enhance the inference performance.

**Strengths:**

- It 's generally an interesting and comprehensive study.

The authors provide an integrated defect benchmark with fine-grained annotations, as well as a diffusion-based defect generation method to augment the dataset.
- The methodology is generally clarified and easy to follow.

Both the dataset reconstruction process and the technical details of the data generation method are clearly explained.

- An integrated and refined dataset is beneficial for the defect segmentation research community.

**Weaknesses:**

- unclear motivation of the choice of the base benchmarks
- insufficient experiments

**Questions:**

My primary concerns are focused on the experiment section.

Firstly, I have some questions regarding the quantitative evaluation results of multiple defect segmentation methods across the different Defect Spectrum reannotated datasets presented in Table 3. The table shows that the average performance of these methods is quite similar, despite their varying performance on individual defect image types. This raises doubts about the suitability of this benchmark for assessing the discrimination ability of different models.

Secondly, there are limitations to the quantitative evaluation of the generation quality of the proposed data generation method. In Table 4 of section 4.4, only the performance of the best baseline is reported, which may not be sufficiently persuasive.

Furthermore, I am curious about the reasons behind choosing to integrate and reannotate those four datasets from the diverse benchmarks listed in Table 1.

---

> ### Author Response · Authors · 2023-11-18
>
> Thank you for your insightful query regarding our paper, we will address your concerns one by one:
>
> **Concerns on the quantitative evaluation result**: The concern about the similar average performance of various methods across different datasets, despite their varying performance on individual defect image types, is a valid point. However, if we look at the mean IOU report in Table 3, the min and max MIOU reported are 45.40 and 51.58. The segmentation results on the CityScapes dataset across various baselines also demonstrated similar MIOU results between 76.9 to 81.6, except for the traditional UNet(69.1).  As for the varying performance on individual classes, this is due to the different granularity that exists in different defect types, making some of the defective regions comparatively smaller than other types.
>
> **Extended experiments on other baselines**: The decision to select the best baseline aims to demonstrate Defect-Gen's capability in the most demanding scenarios, particularly where baseline performance is already high. This approach highlights the robustness and effectiveness of our method, even under challenging conditions. To further underscore the efficacy of Defect-Gen, we conducted comprehensive experiments using three distinct segmentation models: Mask2Former, MIT-B0, and DeepLabV3+, as detailed in the table below (also see **Appendix Section A** in the updated paper). The results from these models reinforce our findings, demonstrating that Defect-Gen not only excels with top-performing methods but also significantly enhances the performance of less advanced models. This broad applicability underscores the versatility and practical utility of our approach.
>
> Performance (mIoU) comparison between models trained with and without synthetic data. The bolded text indicates results with synthetic data.
> | Model        | MVTec                     | VISION                    | Cotton                     |
> |--------------|---------------------------|---------------------------|----------------------------|
> | DeepLabV3+   | 51.58/**55.55**           | 52.33/**53.46**           | 48.73/**58.58**            |
> | Mask2Former  | 45.70/**50.16**           | 54.12/**55.47**           | 64.09/**65.39**            |
> | MiT-B0       | 46.45/**56.21**           | 49.62/**50.75**           | 50.52/**55.86**            |
>
> **The reason to choose MVTEC, VISION, DAGM, and Cotton datasets**:
> **MVTEC Dataset**: Renowned for its high quality, the MVTEC dataset is a well-established benchmark in the field. Its high-resolution images and large-scale nature provide a robust testbed for models, enabling detailed defect detection.
>
> **VISION Dataset**: as the largest scale dataset among those we evaluated, VISION has high variance and high-resolution images. This dataset challenges models with a wide array of defect types and imaging conditions, making it an excellent benchmark for assessing a model's versatility.
>
> **DAGM Dataset**: Unique in its use of special imaging techniques, DAGM provides images that are far from natural representations. This characteristic is crucial for testing a model's ability to generalize and perform in less conventional or more challenging imaging environments.
>
> **Cotton Dataset**: As a real-world industry dataset produced directly in factory settings, the Cotton dataset offers high variance and practical relevance. It simulates real-world industrial conditions, thereby providing a realistic assessment of a model's performance in actual deployment scenarios.
>
> Finally, I sincerely appreciate the valuable insights provided by the reviewers and warmly welcome any further criticisms or suggestions. Your timely feedback is crucial for the enhancement of this paper. I look forward to your valuable comments.

---

> ### Author Response · Authors · 2023-11-21
>
> Dear reviewer J4Kk,
>
> As the window for reviewer-author interaction is closing soon, on November 21st, I wanted to extend my sincerest gratitude for the invaluable time and effort you have dedicated to reviewing our work. To ensure that we have met your expectations, may I kindly ask if you find our responses satisfactory and if there are any remaining issues that need further clarification or improvement?

---

> > ### Comment · Reviewer_J4Kk · 2023-11-21
> >
> > I thank the authors for the detailed responses. They have addressed my concerns, and as a result, I have decided to maintain my initial rating.

---

### Meta-Review · Area_Chair_Htjo · 2023-12-05

**Metareview:**

After discussion, major concerns about the dataset solution, comprehensive analysis and  comparison over previous works remain. After carefully reading the paper, the review comments, the AC deemed that the paper should undergo a major revision, thus is not ready for publication in the current form.

**Justification For Why Not Higher Score:**

N/A

**Justification For Why Not Lower Score:**

N/A

---

### Decision · Program_Chairs · 2024-01-16

Reject